# On Average Losses of Low-Frequency Sound in a Two-Dimensional Shallow-Water Random Waveguide

**Oleg E. Gulin *** and **Igor O. Yaroshchuk**

V.I. Il'ichev Pacific Oceanological Institute, Far-East Branch of Russian Academy of Sciences, 690041 Vladivostok, Russia; yaroshchuk@poi.dvo.ru

**\*** Correspondence: gulinoe@poi.dvo.ru; Tel.: +7-423-231-2617

**Abstract:** For a low-frequency sound signal propagating in a two-dimensionally inhomogeneous shallow-water waveguide, the influence of random bathymetry (fluctuating bottom boundary) was considered based on the local-mode approach and statistical modeling using first-order evolution equations. The study was carried out in shallow sea conditions corresponding to the coastal waveguides of the Russian Arctic seas. Here, a feature was the presence of an almost homogeneous water layer with various characteristics of seabed sediments. To describe the latter, a random model of the impedance was adopted. For the conditions of a strongly penetrable bottom boundary, on average, the calculations predicted adequate weak effects of bathymetry fluctuations on the average sound intensity compared to the effect of fluctuations in the sediment parameters and volumetric random inhomogeneities of the water column. In addition, it was shown that, in terms of statistics, the roughness of the bottom boundary perturbed the average sound intensity in a shallow-water waveguide differently than volumetric fluctuations in the speed of sound. The dependence of the statistical effects (the first and second moments of the signal intensity) on the parameters of the waveguide and the frequency range was studied. As a result of numerical modeling, comparative quantitative estimates of the influence of both the random roughness of the bottom interface and fluctuations of bottom sediment parameters on the average losses of the propagating signal, not presented in the literature, were obtained.

**Keywords:** shallow-water acoustics; range-dependent waveguides; local modes; randomly inhomogeneous impedance of the bottom; rough bottom boundary; statistical modeling





## 1. Introduction

In this paper, the combined effect of random bottom inhomogeneities, its rough surface, and fluctuations in liquid sediment parameters (impedance) on energy losses in the course of the propagation of low-frequency acoustic signals in a two-dimensional shallow-sea waveguide was considered. A variety of works have been devoted to the scattering of sound on rough surfaces, the most famous of which are listed in references [1–12]. They outline the main approaches to an approximate theoretical analysis of the problem. The most common analytical methods are the perturbation theory, the Kirchhoff method, and the integral equation method. Most researchers use semi-analytical approaches, in which an approximate analytical model of wave scattering is first proposed, and in the next stage, specific numerical calculations are carried out for it. As emphasized in review [10], any complication in the scattering model is accompanied by a loss in the visibility of interpretations and an increase in difficulties in the numerical simulation of the scattered field. Early theoretical studies treated surface and volume scattering as completely different problems. However, such an assumption in the acoustics of a shallow sea often does not correspond to practice. Thus, for example, it is experimentally impossible to distinguish scattering from rough seabed interfaces and scattering from volumetric inhomogeneities of

sediments. For this reason, an increasing number of authors have recently been studying these two types of sound scattering simultaneously (for example, [11,12]).

To calculate the scattered field from both volumetric and surface inhomogeneities of the medium, the vast majority of researchers use the ray method, the parabolic equation method (MPE), and the modal approach [5,8,13]. As is known, the ray method is only applicable for high-frequency fields. It is also burdened with computational difficulties at the regions of ray turning points and caustics. The MPE also experiences difficulties in describing the areas of sound focusing in a waveguide, in the case of long-range propagation with diffraction by inhomogeneities [14], and also with a strict formulation of the boundary conditions for waveguides with irregular interfaces [13,15].

Works regarding modal waveguides in the low-frequency range that are close to our study in the formulation of the problem are given in [6,16–18]. So, in [16], the Kirchhoff method for Green's function, first developed in [1–3,6], was used in the form of a mode representation for layer-wise inhomogeneous waveguides with flat boundaries, and the scattering on the roughness of the boundaries (bottom and surface) was calculated using the perturbation method. For small distances of sound propagation, the correlation functions of the field in the vertical and horizontal directions were calculated. Additionally, the coherence between the regular and scattered parts of the sound field was calculated. In [17,18], the local-mode approach, proposed earlier in [19,20], as well as the assumption of convolution along the horizontal coordinate of the adiabatic solution for a 2D waveguide with mode-coupling coefficients, was used to calculate scattering using the random roughness of the seabed interface. The authors considered the scattering of broadband signals at a fixed distance using the Fourier method. A comparison was made with the results of the Born approximation, which was recognized as adequate in the simple Pekeris waveguide model considered. The dependence of propagation losses on the properties of a randomly inhomogeneous waveguide was not studied in [16–20].

With regard to the effect of non-stratified fluctuations in the parameters of bottom sediments, primarily the speed of sound, on the propagation of sound, there are very few such works. The influence of the 2D inhomogeneities of bottom sediments on the transmission losses of low-frequency sound in a deterministic formulation was considered in [21,22], as applied to the Russian shelf of the Arctic seas (the Kara Sea). In the statistical formulation, for 2D sound velocity fluctuations in bottom sediments, studies were performed in [23,24], wherein the possibility of a significant effect of the random character of sediment parameters on transmission losses of low-frequency sound was shown.

As established in [25,26], the greatest perturbing effect on the sound intensity is achieved when the bottom boundary of a shallow-water waveguide is, on average, highly penetrable. In such a situation, sound velocity fluctuations in the water layer can slow down the decay of the average intensity (transmission loss) by tens of decibels in relatively short distances, which are relevant for field studies in a shallow sea. Fluctuations in the parameters of the sedimentary bottom layer lead to a similar effect of the attenuation of the average transmission loss of low-frequency sound in water [24]. The degree of manifestation of these effects in a shallow-water waveguide with a highly penetrable bottom interface is mainly determined by the horizontal scales of parameter fluctuations [27]: the larger these scales, the stronger the effects. It is of obvious interest to compare the effect of both types of random inhomogeneities (volumetric fluctuations in the speed of sound and the roughness of a bottom interface), present in the real environment of a shallow sea, on the propagation of low-frequency sound signals. It is important to find out the features in the behavior of the average energy characteristics of a sound signal when it is perturbed by volumetric and surface inhomogeneities, to obtain quantitative estimates of the effect of inhomogeneities, and to analyze the dependence on parameters.

The present study was carried out on the basis of statistical modeling [24,28,29] of the average sound intensity and its fluctuations, which describe energy losses and scintillations in the course of signal propagation in a randomly inhomogeneous medium of a shallow sea. The solution for individual random realizations of the parameters was obtained using the

universal local-mode approach developed in [30–34]. In the framework of this approach, which is suitable for studying a wide class of inhomogeneities in a shallow sea, the mode amplitudes were generally sought based on the reformulation of the original boundary value problem into first-order causal equations.

## 2. Mathematical Statement of the Problem and Some Analytics

The acoustic field of frequency $\omega$ in a two-dimensionally inhomogeneous waveguide of a shallow sea is described by linear acoustic equations with boundary conditions on the surface and bottom interface of the waveguide. In the axially symmetric formulation of the problem, in the presence of a variable density $\rho$ in the medium (water column and liquid bottom sediments), for the acoustic pressure function $p$, the equations of linear acoustics are reduced to an equation of the form [13,15]

$$\rho r^{-1} \frac{\partial}{\partial r}\left(r\rho^{-1}\frac{\partial p}{\partial r}\right) + \rho\frac{\partial}{\partial z}\left(\rho^{-1}\frac{\partial p}{\partial z}\right) + \frac{\omega^2}{c^2}p(r,z) = -\frac{\delta(r)\delta(z-z_0)}{2\pi r} \tag{1}$$

where $(r,z)$ are the coordinates of the cylindrical system, and the point source of radiation is located at the point $(r = 0, z = z_0)$; $c$ is the speed of sound in water. The boundary condition on the surface $p(r,0) = 0$, and the condition on the bottom corresponds to the continuity of the pressure and the velocity component normal to the boundary $H(r)$. It is assumed that the field radiation conditions are satisfied at infinity $z \to \infty$. In the horizontal direction $r$, both continuity conditions and radiation conditions are also implied. In the wave zone of the source, the pressure field $p(r,z)$ is sought using the expansion in local modes of an irregular 2D waveguide:

$$p(r,z) = \sum_m G_m(r)\,\varphi_m(r,z)\,;\ \ \rho\frac{\partial}{\partial z}\left(\rho^{-1}\frac{\partial}{\partial z}\varphi_m(r,z)\right) + \left[k^2 - \kappa_m^2(r)\right]\varphi_m(r,z) = 0 \tag{2}$$

In Equation (2), $k = \omega/c$, $\kappa_m(r)$ are the eigenvalues, and $\varphi_m$ are the eigenfunctions of the Sturm–Liouville problem ($m = 1, 2 \dots$ ), which, on the surface and at the bottom of the ocean, satisfy the following boundary conditions: $\varphi_m(r,0) = 0$, $\varphi_m(r,H) + g_m(r)\varphi'_m(r,H) = 0$, $\varphi'_m(r,H) = (\partial\varphi_m(r,z)/\partial z)|_{z=H}$. Here, $g_m(r)$ characterizes the impedance of the penetrable bottom and, together with the rough boundary $H(r)$, it is a random function due to fluctuations in the sound speed $c_1$ within the seabed (if necessary, without changing the formulation of the problem, one can also consider density fluctuations $\rho_1$). From Equation (2), it is obvious that the eigenfunctions and eigenvalues, as well as the local modes of the waveguide, will be random functions of $r$. Previously, it was shown [24–27] that in an irregular waveguide, in the forward-scattering approximation, the modal amplitudes $G_m(r)$ are determined by the following analytical form of the solution ($\kappa_m r \gg 1$):

$$G(r) = \{G_m(r)\} = A(r)\exp\left\{\int_0^r \left[i\kappa(\xi) - \left(\kappa(\xi)V(\xi)\kappa^{-1}(\xi) - V^T(\xi)\right)/2\right]d\xi\right\}b(0), \tag{3}$$

where $\kappa(r)$ is the diagonal matrix of eigenvalues $\{\kappa_m(r)\}$, $A(r) = (i/8\pi r)^{1/2}\kappa^{-1/2}(r)\kappa^{-1/2}(0)$, $b(0) = \{\varphi_m(0,z_0)\kappa_m^{1/2}(0)\}$ is the column vector of the initial amplitudes of modes, and $\exp\{...\}$ is the matrix exponential. $V(r)$ is a matrix with elements $V_{mn}(r) = \int_0^\infty \frac{\varphi_m(r,z)}{\rho(r,z)}\frac{\partial\varphi_n(r,z)}{\partial r}dz$, and $V^T(r)$ is a transposed matrix $V$. The latter matrices describe mode coupling due to horizontal changes caused by fluctuations of the sound speed within the seabed and the random roughness of the bottom interface boundary. It is important to note that the original formulation of the boundary condition at the bottom for Equation (1) implies the continuity of the velocity component normal to the local area of the bottom interface, while in the framework of the method of local modes, the vertical velocity component of the modes is continuous in the local area of a rough bottom interface in the following way:

$\rho^{-1}\varphi'_m(r,H-0) = \rho_1^{-1}\varphi'_m(r,H+0)$. Therefore, in order to satisfy the required continuity condition for the velocity component normal to the boundary $H(r)$, Expression (3) must involve the matrix $V^T(r) = -V(r) - \int_0^\infty \varphi_m(r,z)\varphi_n(r,z)\frac{\partial}{\partial r}\left(\frac{1}{\rho(r,z)}\right)dz$ [15,32]. It is this matrix that provides the correct account for continuous variations in the interface $H(r)$ with a jump-like change in density when passing through this interface. Naturally, if, along with density jumps at the interfaces, there are continuous changes in density in the medium, they are also taken into account by the matrix $V^T(r)$. Let us give expressions for the eigenvalues $\kappa_m(r)$ and matrix $V(r)$ that are useful in analysis and calculations. For a continuous interface $H(r)$, in the presence of inhomogeneities $c_1(r,z)$ in the sediment half-space, the following expressions [15,27] are valid:

$$\frac{\partial\kappa_m^2(r)}{\partial r} = \left(\frac{1}{\rho_1}-\frac{1}{\rho}\right)\frac{\partial H(r)}{\partial r}\left\{\left(\frac{\partial\varphi_m(r,z)}{\partial z}\Big|_{z=H}\right)^2 - [k_1^2(r,H)-\kappa_m^2(r)]\varphi_m^2(r,H)\right\}+$$

$$+\frac{(k^2-k_1^2(r,H))\varphi_m^2(r,H)}{\rho_1}\frac{\partial H(r)}{\partial r} + \int_{H(r)}^\infty \frac{\varphi_m^2(r,z)}{\rho_1}\frac{\partial k_1^2(r,z)}{\partial r}dz\ ; \tag{4a}$$

$$V_{mn}(r) = [\kappa_n^2(r)-\kappa_m^2(r)]^{-1}\left(\frac{1}{\rho_1}-\frac{1}{\rho}\right)\frac{\partial H(r)}{\partial r}\left\{\frac{\partial\varphi_m(r,H)}{\partial z}\frac{\partial\varphi_n(r,H)}{\partial z}-[k_1^2(r,H)-\right.$$

$$\left.-\kappa_m^2]\varphi_m(r,H)\varphi_n(r,H)\right\} + \frac{(k^2-k_1^2(r,H))\varphi_m(r,H)\varphi_n(r,H)}{[\kappa_n^2(r)-\kappa_m^2(r)]\ \rho_1}\frac{\partial H(r)}{\partial r} + \tag{4b}$$

$$+\int_{H(r)}^\infty \frac{\varphi_m(r,z)\varphi_n(r,z)}{[\kappa_n^2(r)-\kappa_m^2(r)]\ \rho_1}\frac{\partial k_1^2(r,z)}{\partial r}dz.$$

In the first approximation, if we put on the right sides of Equations (4a) and (4b) the eigenvalues and eigenfunctions of the unperturbed waveguide, $\kappa_m(r) \approx \kappa_{0m}$, $\varphi_m(r,H) \approx \varphi_{0m}(H)$, the upper equation for $\kappa_m(r)$ with the initial condition $\kappa_{0m}$ can be integrated over $r$. Then, it can be seen from (4a) that $\kappa_m(r)$ are determined not by the shape of the interface, but by the local depth of the waveguide $H(r)$, by the density jump $\left(\frac{1}{\rho_1}-\frac{1}{\rho}\right)$ at the interface $H(r)$, and also by the perturbation of the sound speed $\delta c_1$ in the sediment half-space: $k_1 = \omega/c_1(r,z)$, $c_1(r,z) = \langle c_1\rangle + \delta c_1(r,z)$, $\langle c_1\rangle$ is the mean value. At the same time, the elements of the coupling matrices $V(r)$ in (4b) depend on the local slopes of the interface $\partial H(r)/\partial r$, on the derivatives of sound velocity perturbations $\partial[\delta c_1(r,z)]/\partial r$, and on the density difference when passing the interface boundary. In addition, a well-known fact in (4b) is the inverse proportionality $V_{mn}(r) \sim [\kappa_n(r)-\kappa_m(r)]^{-1}$. Although Equations (4a) and (4b) are exact and allow one to perform a useful qualitative analysis of the influence of inhomogeneities within the framework of the perturbation method, quantitatively, this approximation is of little use for the waveguide models with a highly penetrable bottom interface studied in this work. Therefore, for numerical simulation, precise calculations of both the modal wave numbers $\kappa_m(r)$ and the elements $V_{mn}(r)$ of the coupling matrices were performed according to the algorithms of the authors [24,26,28,30,34]. When performing the calculations, we were guided by the criterion of the smoothness of random variations in the local slopes $\partial H(r)/\partial r \ll 1$.

Expression (3) takes into account the scattering of modes at any angles not exceeding 90°. Further in the work, we call this the OW (one-way propagation) solution [13]. Backscattering can also be taken into account within the approach used [30–34]. However, in the problems of low-frequency sound propagation in the sea, the role of backscattering is negligible [15,33]. Therefore, in order to not complicate the study with irrelevant details, backscattering was not considered here.

If both the density of the medium and boundaries do not change in the horizontal direction, then the matrix $V(r)$ in (3) transforms to skew-symmetric: $V_{mn}(r) = -V_{nm}(r)$, $V_{nn} = 0$. If the inhomogeneities of the medium change smoothly and the scattering angles are small, so that $\kappa(r)V(r)\kappa^{-1}(r) \approx V(r)$, then the WKB approximation in the horizontal

direction can be obtained from Equation (3) [8,22], as well as the approximation of the MPE [30,35]. Often, these WKB and MPE methods for waveguide models with horizontal boundaries and constant (in *r*) density provide a good approximation to the true OW. However, for waveguides with variable densities and the rough bottom boundary studied in this work, these approximations, as well as the adiabatic approximation ($V_{mn} = 0$), do not strictly satisfy the boundary condition on the non-planar interface $H(r)$.

By calculating the pressure field $p(r,z)$ according to Equations (2) and (3) for each random realization $c_1(z,r)$, $H(r)$ from an ensemble of *N* realizations, it is easy to obtain the change in the average intensity or the average function of transmission loss for sound propagating along the path in a randomly inhomogeneous waveguide:

$$\langle I \rangle = \left\langle |p|^2 \right\rangle = \sum_n \left\langle |G_n|^2 \, |\varphi_n|^2 \right\rangle + \sum_{(n \neq m)} \left\langle G_n \, G_m^* (\varphi_n \, \varphi_m^*) \right\rangle. \tag{5}$$

In Equation (5), angled brackets mean statistical averaging, which is replaced by algebraic formulas in calculations. The first sum of modes in Equation (5) represents the incoherent terms and describes the averaged (over the scale of interference) intensity decay law in the waveguide. The second sum of coherent terms describes the wave interference structure of the sound field, which is superimposed on a smooth averaged law of decay. With statistical averaging, the contribution to the intensity of the sum of coherent terms at low frequencies decreases rapidly with distance. However, for multimode situations, which take place, for example, in the high-frequency range and with weak attenuation in the waveguide of low-number modes, the oscillatory intensity structure Equation (5) can also be noticeable at distances of tens of kilometers. In this case, to estimate the effect of inhomogeneities on the levels of transmission loss, it is rational to only take into account the first sum of incoherent terms in Equation (5), which is carried out later for a signal with a frequency of 500 Hz.

Similarly to Equation (5), according to well-known relations, other statistical characteristics of the intensity of interest can be calculated. For example, an important indicator of sound intensity in a randomly inhomogeneous waveguide is the scintillation index $S^2$, where $S = (\langle I^2 \rangle - \langle I \rangle^2)^{1/2} / \langle I \rangle$ [7,24,36].

## 3. Stochastic Waveguide Model

A shallow-water waveguide was considered, consisting of a water layer and a bottom in the form of a half-space of liquid sediments. The water–sediment interface randomly fluctuated (it was a statistically rough surface). A tone sound signal of frequencies 250 and 500 Hz propagated in the waveguide. To carry out a numerical analysis, in accordance with Equations (2), (3), and (5) of Section 2, a reference was made to the values of parameters that are typical for the Russian shelf zone of the Arctic seas, in particular, the Kara Sea [8,21,22]. The waveguide had an average depth $\langle H(r) \rangle = 40$ m, a horizontal surface, and a rough boundary of the bottom. In the water layer, there were uniform profiles of sound speed $c = 1460$ m/s and density $\rho = 1$ g/cm$^3$. The seabed consisting of unconsolidated sediments was modeled by an absorbing liquid half-space with a refractive index at the water–bottom interface $n = (c/c_1)(1 + i\beta_1)$, $\beta_1 = 0.02$. In bottom sediments, following the measurement data given in [21,22], we set the impedance using the density, $\rho_1(r) = \langle \rho_1 \rangle = 1.85$ g·cm$^{-3}$, and the speed of sound $c_1(r,z)$, which randomly varied along the propagation path of a signal. We also took into account the fact that random variations in density $\delta\rho_1(r)$, $\rho_1(r) = \langle \rho_1 \rangle + \delta\rho_1(r)$, have a much weaker effect on sound propagation than fluctuations in the speed of sound. This fact is well known from the theory (see, for example, [5,8,13,24]). The variations in density in bottom sediments can be neglected if not-too-low radiation frequencies are examined ($f = 2\pi\omega > 1$ Hz) and there are no large-amplitude jumps of $|\delta\rho_1/\langle \rho_1 \rangle|$ in the liquid sediments. Random field $c_1(r,z) = \langle c_1 \rangle + \delta c_1(r,z)$ assumed Gaussian fluctuations with an exponential correlation function: $B_{c1}(r_2 - r_1, z_2 - z_1) = \sigma_{c1}^2 \exp(-|r_2 - r_1|/L_{rc} - |z_2 - z_1|/L_{zc})$. Gaussian fluctuations of the rough water–sediment interface were set similarly: $H(r) = \langle H \rangle + \delta h(r)$, $B_h(r_2 - r_1) = \sigma_h^2 \exp(-|r_2 - r_1|/L_h)$. Thus, the

stochastic waveguide studied further in this work was completely specified by the first and second statistical moments of the main fluctuating parameters: $c_1(r,z)$, $\rho_1(r)$ and $H(r)$. In Figure 1a,b, as an example, a graphical illustration of a stochastic waveguide model is shown. This model is presented for several arbitrary realizations (from the statistical ensemble of realizations) of random bathymetry and sound speeds at the water–sediment interface for the scale of $L_{rc} = L_h = 1$ km.

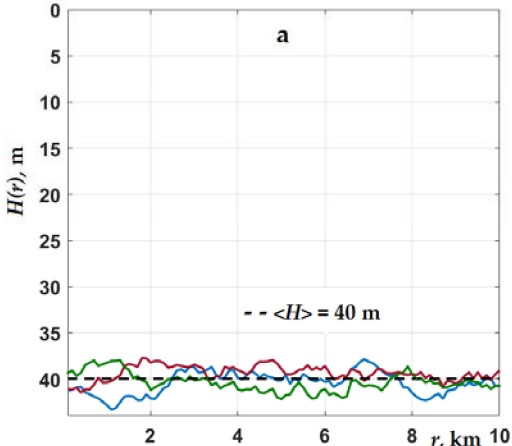
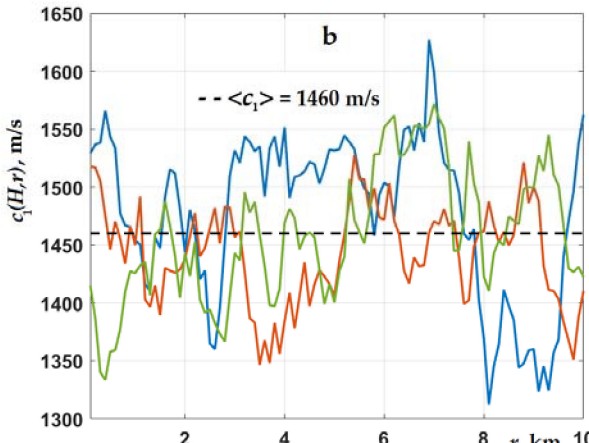

**Figure 1.** Illustration of a stochastic waveguide model. (**a**)—three random realizations of waveguide bathymetry fluctuations; (**b**)—three random realizations of fluctuations in the speed of sound $c_1$ at the interface $z = H(r)$. $L_{rc} = L_h = 1$ km.

The impedance function $g_m(r)$ in the boundary condition to Equation (2) was determined by its local values in the cross sections of the comparison waveguides. For homogeneous stratifications of the sound speed and density both in water and sediments, the comparison waveguides were Pekeris waveguides. Note that the choice of the exponential form of the correlation functions $B_{c1}$ and $B_h$ was dictated by convenience. The results of statistical analysis were affected not by the type of functions, but by the characteristic scales of inhomogeneities $L_{rc}$, $L_h$ [26].

### 4. Results of the Propagation Loss Statistical Analysis

The statistical modeling of average intensity (4) was performed for two scenarios of a shallow-water waveguide. For the first scenario, the water-bottom interface was highly penetrable: $\langle c_1 \rangle = c = 1460$ m/s. For the second scenario, it had a certain degree of rigidity: $\langle c_1 \rangle = 1500$ m/s. In the first case, as was shown in [25,26] for volumetric sound velocity fluctuations in the water layer, as well as for fluctuations of the impedance $g_m(r)$ in [23,24], the maximum statistical effect was achieved in the course of sound signal propagation. Based on the processing data of works [21,22], the characteristic scale of inhomogeneities $L_{rc}$ was chosen to be 1 km, $L_{zc} = 30$ m ($L_{zc} \gg \lambda$, where $\lambda$ is the sound wavelength), and the intensity of fluctuations $\sigma_{c1}^2 = \langle (\delta c_1 / \langle c_1 \rangle)^2 \rangle = 1.7 \cdot 10^{-3}$ (corresponds to $|\delta c_1| \approx 60$ m/s). The characteristic scale of change in $H(r)$ was considered to be $L_h = 100$ m [16] and 1 km [36], and the intensity of fluctuations $\sigma_h^2 = \langle (\delta h)^2 \rangle = 1$ m$^2$. In the process of numerical simulation, to obtain a reliable statistical result by averaging, an ensemble of realizations $N = 10^3$ was used. Below are the numerical results of the statistical modeling of the intensity for the specified waveguide scenarios in the presence of interface boundary roughness and sound velocity $c_1$ fluctuations in the underlying liquid sediments, which means that the bottom impedance was random. Throughout the graphs, transmission loss is given in decibels relative to the level at a distance of 1 m from the source.

Let us start the analysis with the first waveguide scenario, when the water–bottom interface was highly penetrable, $\langle c_1 \rangle = c$. In such a waveguide, a signal propagating along the path passed bottom sections with inhomogeneities of the acoustically "soft" ($c_1 < c$)

and "rigid" ($c_1 > c$) types [24] (see Figure 1b), which is typical for the sea shelf with gas saturation in bottom sediments [21,37]. One example of this is the Arctic shelf zones, whose bottom sediments are known to be characterized by an increased content of gas with random spatial distribution. In the case when $c_1 < c$, even in the absence of absorption $\beta_1$ no propagating modes were excited in the waveguide, and all modes were proven to be leaky. In the second case, for $c_1 > c$, depending on the degree of "rigidity" of the bottom, a number of the first modes were trapped and weakly attenuating (if $\beta_1 \neq 0$). To find the local eigenvalues $\kappa_m(r)$ and eigenfunctions $\varphi_m(r)$, reference to the Pekeris cut on the complex plane of $\kappa_m$ was made. Therefore, the required number of propagating and leaky modes [13] forming a field at distances $r > 100–200$ m was taken into account in the sum (2). As a rule, with the waveguide hydrology and sound frequency described above, it was sufficient for statistical modeling to use 6–10 different types of modes in calculations. From the average intensity curves presented in Figures 2 and 3, it follows that bottom impedance fluctuations, on average, had a much stronger effect on signal transmission losses than the bottom interface roughness (compare curves 2 and 1 with 3). Fluctuations $\delta c_1$, as noted above and shown in [23,24], led to a slower decrease in the average intensity with distance. At a distance of 10 km from the source, as seen in Figure 2 (curve 2), this slowdown was 13–14 dB [24] at the scale $L_{rc} = 1$ km. The slowdown became even more significant as the correlation scale $L_{rc}$ increased. At the same time, fluctuations of the boundary $\delta h(r)$, which, as is known from the literature, scatter the signal, on the contrary, increased the transmission loss. However, this effect in the low-frequency region was proven to be very small, amounting to tenths of a decibel for the inhomogeneity scale $L_h = 1$ km (see the inset in the upper right corner of the graph in Figure 2). Some enhancement of the effect (up to $\approx 1$ dB) to a distance of 10 km was observed for a 10-times smaller correlation scale $L_h = 100$ m (see Figure 3, curves 1 and 2, and the inset in the upper right corner of the graph). In this case, due to the lower smoothness of the interface roughness, the coupling of modes in the water column slightly increased. As a result, the acoustic energy, being pumped into modes with higher numbers, transferred relatively faster from the water to bottom sediments. At a distance of more than 3 km from the source, the interference structure of the intensity was suppressed, and the curves shown in Figures 2 and 3 became quite smooth. The average intensity in the middle part of the water layer began to be formed, mainly by the least-attenuated first mode.

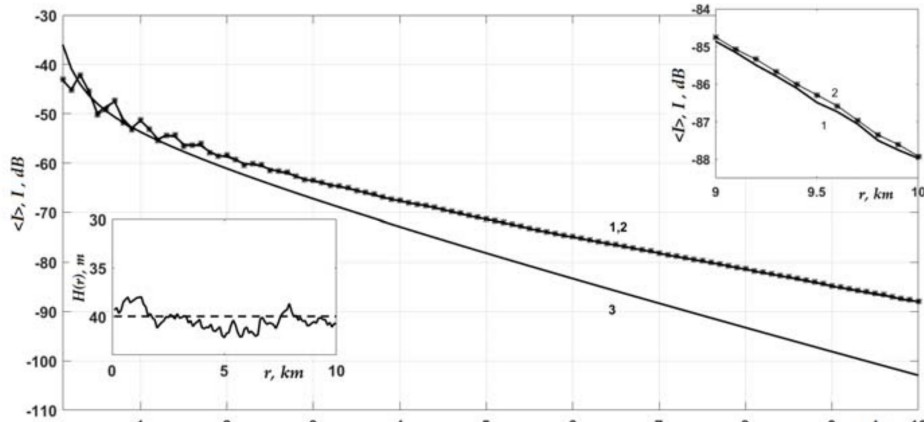

**Figure 2.** Attenuation of the average intensity of a signal with a frequency of 250 Hz in a waveguide with fluctuations both in the impedance of the bottom $\delta c_1$ and the bathymetry $\delta h$. $L_{rc} = L_h = 1$ km; $z = z_0 = 24$ m. Curves: 1 is the OW solutions (2) and (3); 2 is the OW solution for $\delta h = 0$ (markers); 3 is intensity averaged over the interference scale for $\delta c_1 = \delta h = 0$.

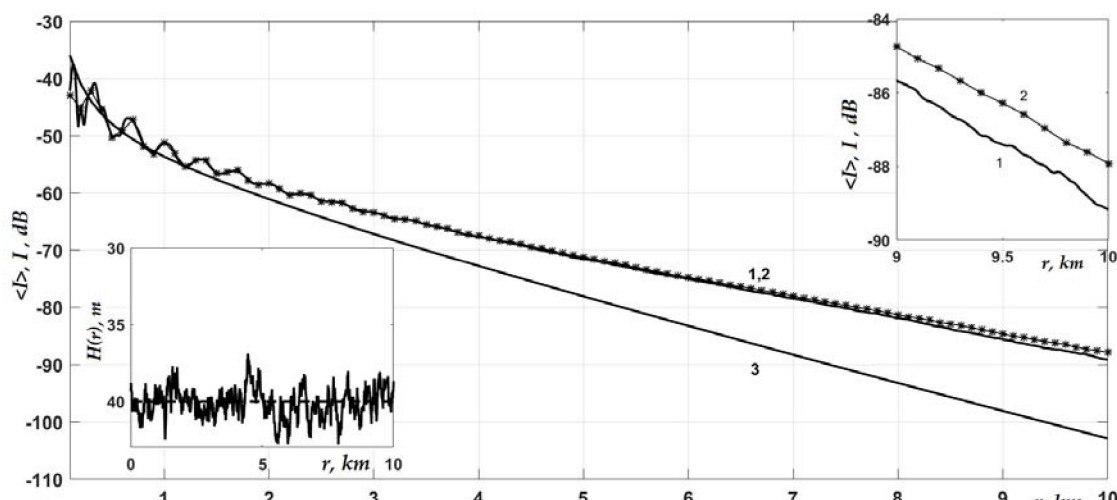

**Figure 3.** The average intensity is similar to Figure 2, but roughness of the bottom interface has a smaller scale, $L_h$ =100 m (see left inset on the graph).

Figure 4 shows a graph of the average intensity when the source was located near the rough boundary. For this observation horizon, the contribution of the first mode to the sound field was weakened (due to a decrease in the amplitudes $\varphi_1(0,z_0)$, $\varphi_1(r,z)$ near the bottom), while modes of higher numbers, $m$ = 2–4, on the contrary, increased. Therefore, qualitatively, the curves shown in Figure 4 oscillated more noticeably than in Figures 2 and 3, but the intensity decreased somewhat faster with distance (compare curves 3 and 5 in Figures 2–4). Quantitatively, the effect of the intensity decay increase on ≈1 dB due to fluctuations of the bottom interface (at $L_h$ = 100 m), mentioned above, was also preserved for the horizon near this interface. This confirms the fact that, as pointed out in [24], the form of solution (3) implies the weak dependence of the magnitude of statistical effects on the horizons of the source and receiver in the waveguide.

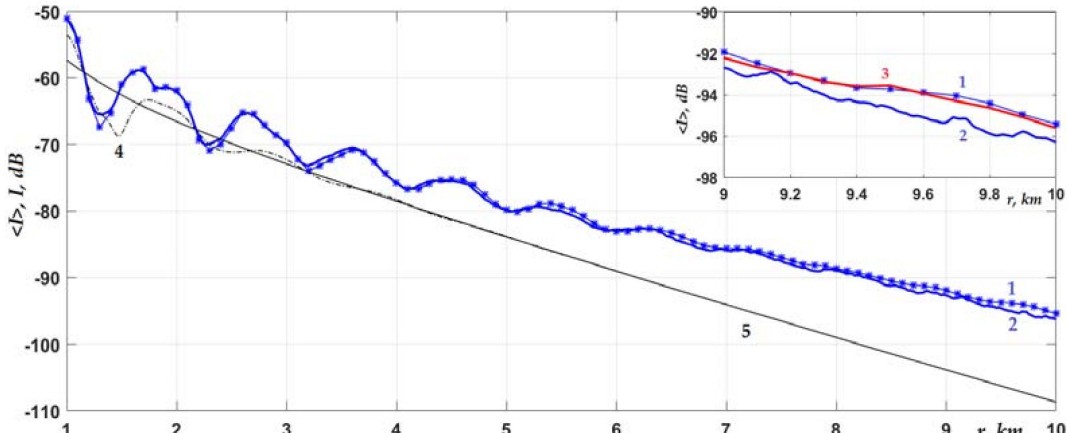

**Figure 4.** Attenuation of the average intensity is similar to Figures 2 and 3 near the bottom. $L_{rc}$ = 1 km; $z = z_0 = 36$ m. Curves: 1 (markers) is OW solutions (2) and (3) for $\delta h = 0$; 2 is the OW solution for $L_h$ = 100 m; 3 is the OW solution for $L_h$ = 1 km; 4 (dots) is intensity in the unperturbed waveguide ($\delta c_1 = \delta h = 0$); 5 is the intensity 4, averaged over the scale of interference.

Let us turn to the consideration of the second scenario. This was a shallow-water waveguide with, on average, a more rigid water–bottom interface: $\langle c_1 \rangle$ = 1500 m/s, $n = 0.97(1 + 0.02i)$. For this case, the effect of sound velocity fluctuations in sediments (impedance) was significantly weakened due to the presence of a larger number of the first weakly attenuated modes. Therefore, for better visualization, the graphs only show the

effects of the random roughness of the interface boundary (impedance fluctuations were excluded from the consideration). Below Figures 5 and 6 show the laws of intensity decay for the radiation and observation horizons studied earlier both in the middle part of the water layer and near the bottom.

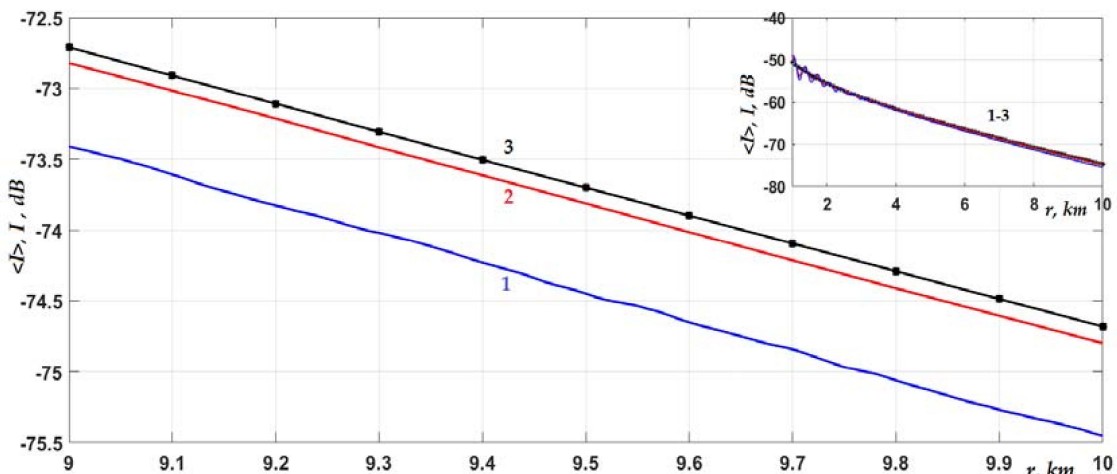

**Figure 5.** Attenuation of $\langle I \rangle$ of a signal with $f = 250$ Hz in the range $r = 9$–10 km in a waveguide with bathymetry fluctuations. $\langle c_1 \rangle = 1500$ m/s, $\delta c_1 = 0$. $z = z_0 = 24$ m. Curves: 1 is OW solution for $L_h = 100$ m; 2 is OW solution for $L_h = 1$ km; 3 (markers) is $I$, averaged over the interference scale ($\delta h = 0$).

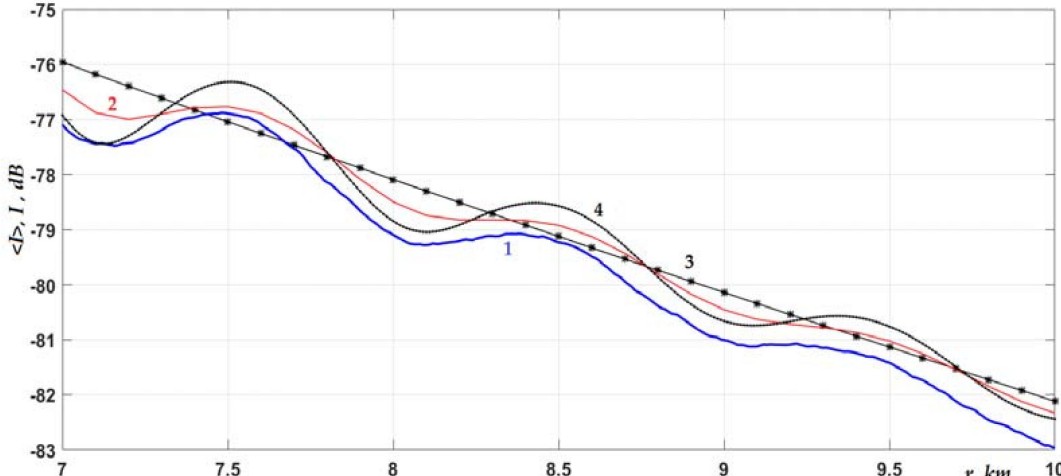

**Figure 6.** Similarly to Figure 5, the average intensity in the range $r = 7$–10 km in the waveguide. $z = z_0 = 36$ m. Curves: 1 is OW solution for $L_h = 100$ m; 2 is OW solution, $L_h = 1$ km; 3 (markers) is $I$ in the unperturbed waveguide ($\delta h = 0$), averaged over the scale of interference; 4 is $I$ in the unperturbed waveguide ($\delta h = 0$).

Figures 5 and 6 show the same features of the average intensity behavior in the waveguide that are established above. These are the weakness of the influence of the random roughness at low frequencies, which did not exceed 1 dB at a distance of 10 km from the source, and the very slow accumulation effect of the influence of inhomogeneities with increases in the propagation distance (compared with the effect on the intensity of a random impedance in Figures 2–4). Additionally, an increase in the influence of interface inhomogeneities with a decrease in the characteristic scale $L_h$ was obvious. In addition, we noted the pronounced oscillatory behavior of the intensity in the course of signal radiation and observation near the interface, where the contribution was significant not only of the first mode $m = 1$, but also of a number of higher modes. This became especially noticeable when the interface ceased to be highly penetrable (the condition $c \approx c_1$ was violated). Thus,

an increase in the "rigidity" of the interface boundary ($c_1 > c$) led to the characteristic wavy dependences shown in Figure 6. An increase in the acoustic "softness" ($c_1 < c$) of the interface boundary led to similar results.

Let us now consider a higher signal frequency $f$ = 500 Hz in a shallow-water waveguide with a rough interface. As is known, with increasing sound frequency, all scattering effects should increase. Therefore, it is important to understand in what range of a frequency one should still expect any significant influence of the random roughness of the interface boundary. As can be seen from Figures 7 and 8, even at a frequency of 500 Hz, perturbations worthy of attention were caused by inhomogeneities of the interface boundary only when the source and receiver were oriented directly near the interface (Figure 8). In this case, the interface boundary itself was not strongly penetrable, but it effectively reflected a number of the first modes (in this case, $m$ = 1–6), which were propagating (trapped) in the waveguide. Therefore, we observed a pronounced interference pattern in the graphs, which caused noticeable differences between the curves at the considered distances from the source. If, however, the condition, $c \approx c_1$, was satisfied at the interface, then the influence of the boundary inhomogeneities would be similar to those shown in Figures 2–4 for a frequency of 250 Hz and would only be determined by the scale of inhomogeneities $L_h$. To obtain adequate conclusions, it was expedient to reduce the interference pattern in Figure 8 to a form free from oscillations by averaging all the curves over the interference scale. This was achieved by only including the incoherent mode sum in Equation (5) in the analysis (see comments under Equation (5)). As can be seen from Figure 9, now, the difference in curves 1 and 2 from curve 3 was within the same range of 0.1–1 dB as for a frequency of 250 Hz (Figures 5 and 6), depending on the scale $L_h$ of the random roughness of the interface boundary. It should be noted that the effect of volumetric fluctuations in the speed of sound (impedance), which manifested itself in slowing down the decay of intensity, increased in the waveguide in proportion to the frequency. Therefore, in the higher-frequency range (500–1000 Hz), sound velocity fluctuations $\delta c_1$ would mask the effect of the interface roughness even more strongly than that shown in the curves in Figures 2–4 for 250 Hz.

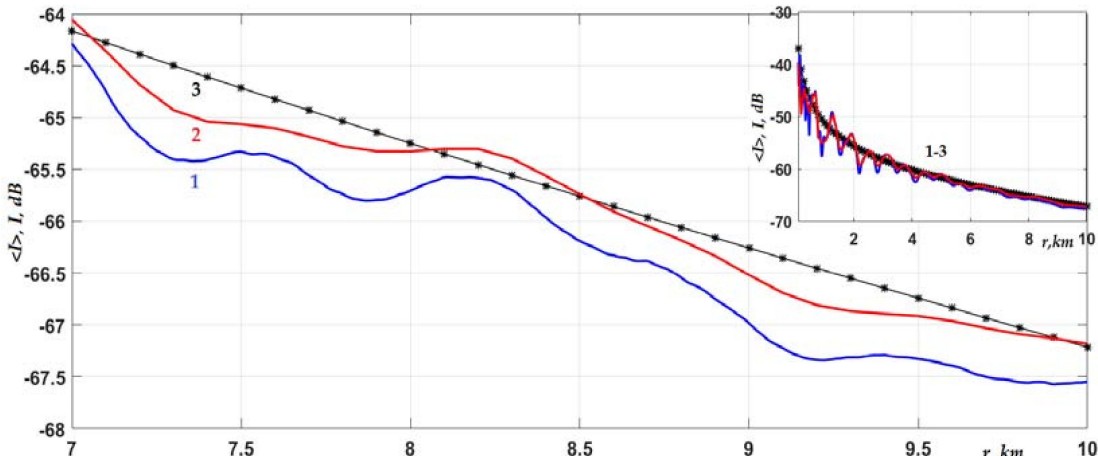

**Figure 7.** $\langle I \rangle$ in the range $r$ = 7–10 km in a random waveguide with bathymetry fluctuations, $f$ = 500 Hz. $\langle c_1 \rangle$ = 1500 m/s, $\delta c_1$ = 0. $z = z_0$ = 24 m. Curves: 1 is OW solution, $L_h$ = 100 m; 2 is OW solution, $L_h$ = 1 km; 3 (markers)—$I$ in the unperturbed waveguide ($\delta h$ = 0), averaged over the interference scale.

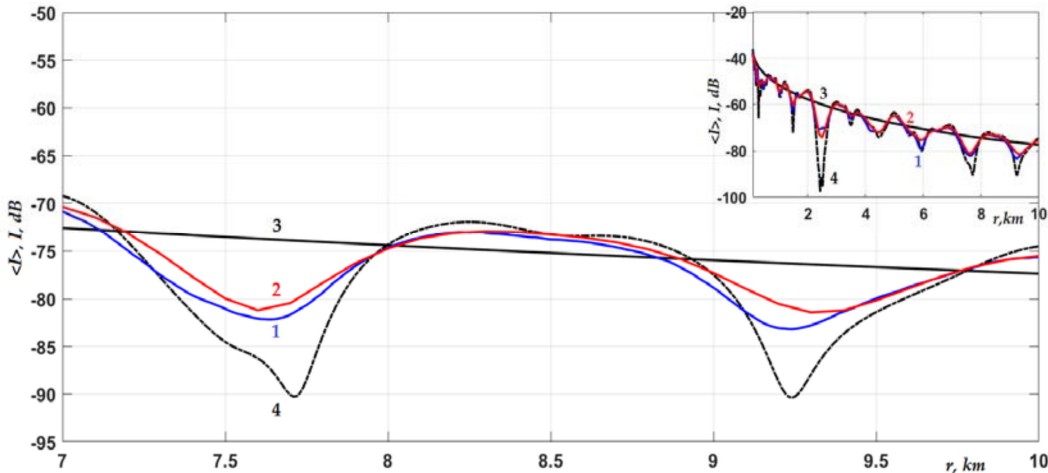

**Figure 8.** Similarly to Figure 7, the average intensity, but near the bottom. $z = z_0 = 36$ m. Curves: 1 is OW solution, $L_h = 100$ m; 2 is OW solution, $L_h = 1$ km; 3 is intensity $I$ in the unperturbed waveguide ($\delta h = 0$), averaged over the interference scale; 4 is intensity $I$ in the unperturbed waveguide ($\delta h = 0$).

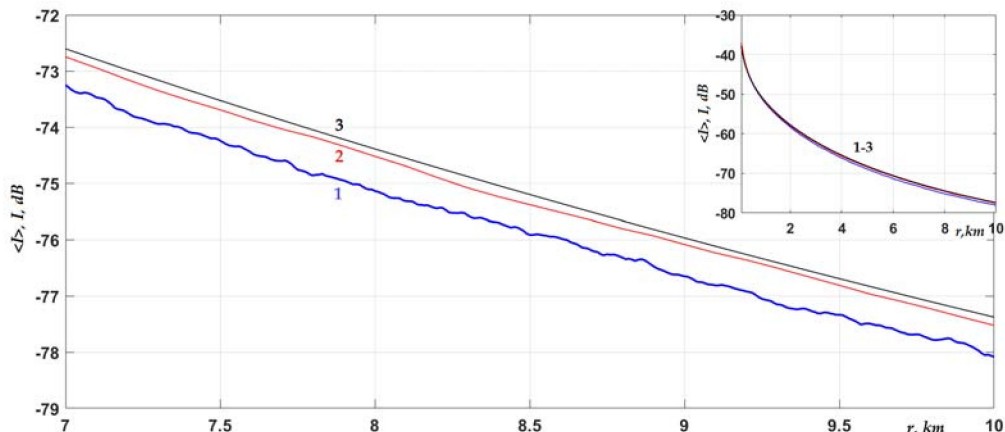

**Figure 9.** Average decay laws for the average intensity in a random waveguide with bathymetry fluctuations, $f = 500$ Hz. $\langle c_1 \rangle = 1500$ m/s, $\delta c_1 = 0$. $z = z_0 = 36$ m. Curves: 1 is OW solution, $L_h = 100$ m; 2 is OW solution, $L_h = 1$ km; 3 is intensity $I$ in the unperturbed waveguide ($\delta h = 0$).

## 5. Scintillation Index Behavior in the Randomly Inhomogeneous Waveguide

As noted above, an important statistical characteristic, in addition to average propagation losses, is the scintillation index $S^2$, which describes sound intensity fluctuations and makes it possible to additionally understand the features of the influence of certain random inhomogeneities on the propagation of an acoustic field in a waveguide. Figure 10 shows the behavior of the scintillation index for the first waveguide scenario, which corresponds to Figures 2–4.

Here, the graphs for different horizons of the source location and the observation point showed an increase in intensity fluctuations along the entire low-frequency signal propagation path. The value of $S^2$ from distances $r > 3$ km exceeded the value 1, which indicated the presence of strong field fluctuations in the waveguide even at rather small distances from the source. In this case, the growth of fluctuations continued without a transition to the saturation regime. Strong fluctuations mean that, at $r > 3$ km, the dominant part of the sound field in the waveguide was formed by a random component. The situation was similar to that which arises in a waveguide in the presence of volumetric sound velocity fluctuations [24–26,35]. The growth of scintillations confirmed that in this case, the main influence was due to random inhomogeneities of the impedance of bottom sediments, and not due to the random roughness of the interface. In this case, the influence of the

random profile $c_1(r,z)$ had a pronounced cumulative character with the signal propagation distance. The last conclusion is confirmed by Figures 11 and 12, in which the scintillation index is presented for a waveguide with a randomly rough interface, but in the absence of sedimentary inhomogeneities. The graphs in Figures 11 and 12 correspond to intensity fluctuations, for which the statistical average is given in Figures 5–8.

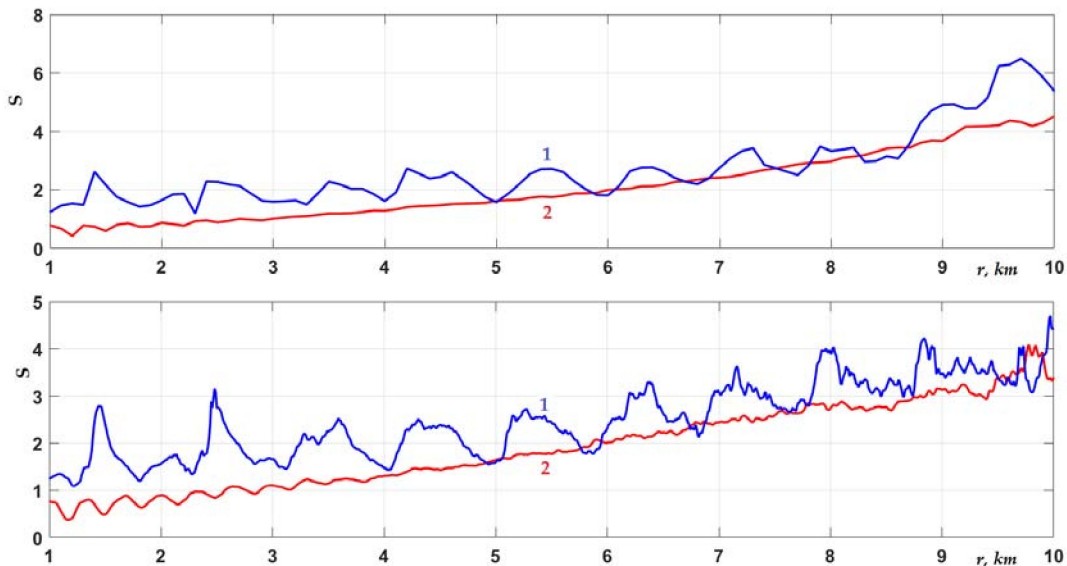

**Figure 10.** Behavior of the scintillation index of a signal of 250 Hz in a waveguide with fluctuations both in the sediments $\delta c_1$ and the bathymetry $\delta h$. $L_{rc} = 1$ km; $c = \langle c_1 \rangle = 1460$ m/s. Top graph corresponds to $L_h = 1$ km; curves: 1 is OW solution, $z = z_0 = 36$ m, 2 is for $z = z_0 = 24$ m. Lower graph corresponds to $L_h = 100$ m; curves: 1 is OW solution, $z = z_0 = 36$ m, 2 is for $z = z_0 = 24$ m.

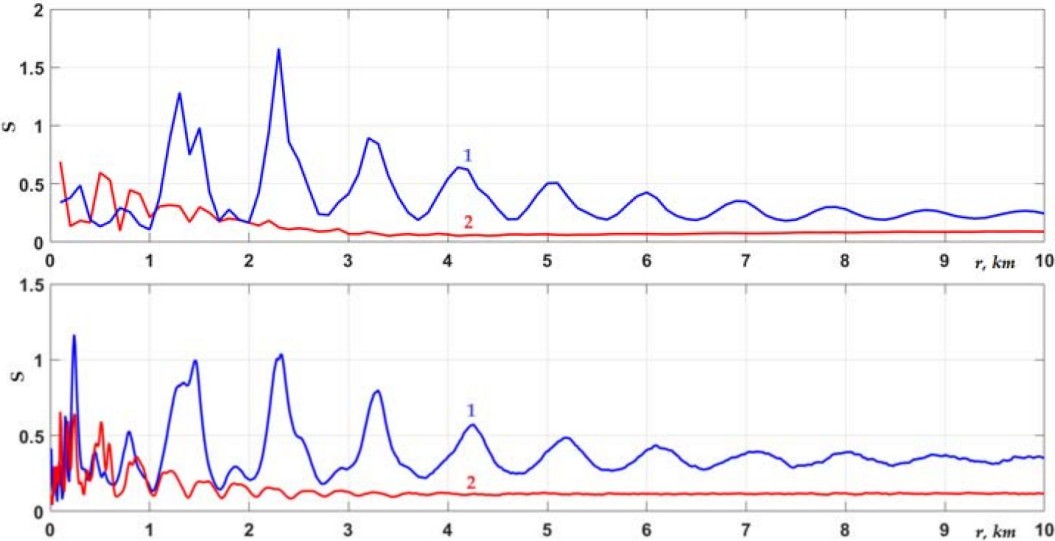

**Figure 11.** Behavior of the scintillation index of a signal of 250 Hz in a waveguide with bathymetry fluctuations $\delta h$. $\langle c_1 \rangle = 1500$ m/s, $\delta c_1 = 0$. Top graph corresponds to $L_h = 1$ km; curves: 1 is OW solution, $z = z_0 = 36$ m, 2 is for $z = z_0 = 24$ m. Lower graph corresponds to $L_h = 100$ m; curves: 1 is OW solution, $z = z_0 = 36$ m, 2 is for $z = z_0 = 24$ m.

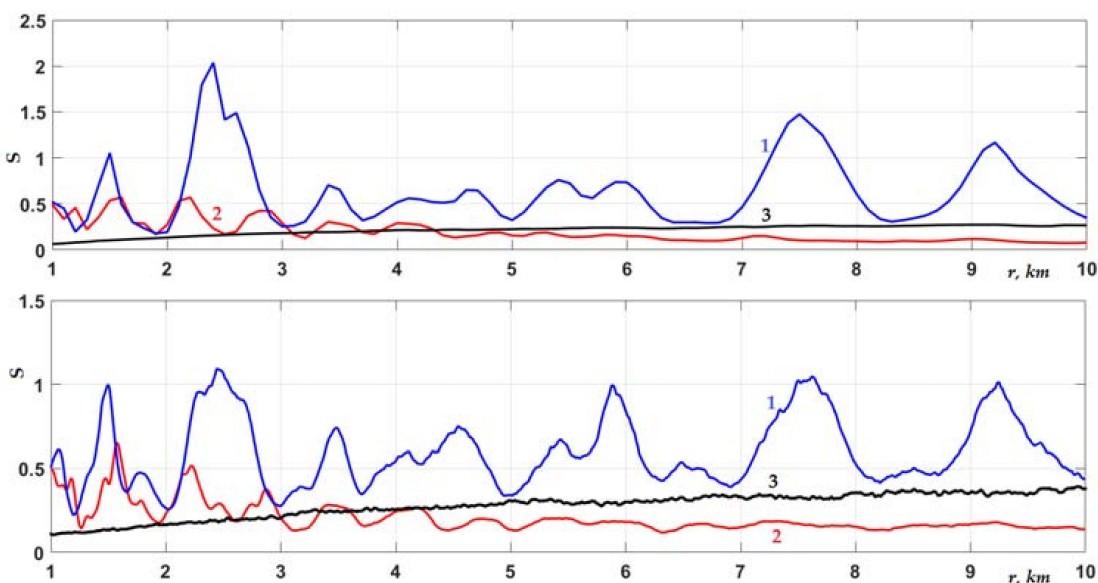

**Figure 12.** S-behavior of a signal of 500 Hz in a waveguide with fluctuations $\delta h$. $\langle c_1 \rangle = 1500$ m/s, $\delta c_1 = 0$. Top graph corresponds to $L_h = 1$ km; curves: 1 is OW solution, $z = z_0 = 36$ m, 2 is for $z = z_0 = 24$ m, 3 is noncoherent sum of modes for $z = z_0 = 36$ m. Lower graph shows similar (to top graph) curves for $L_h = 100$ m.

It is clearly seen that there was no increase in the intensity fluctuations along the path in the absence of sediment inhomogeneities. Now, the scintillation index could only exceed the value of 1 locally, at those distances where minima appeared in the interference pattern of the signal intensity. This is typical for source and receiver horizons located near a rough interface, which is not highly penetrable but has a certain reflectivity. In this situation, a number of first weakly damped modes were excited, which formed the interference pattern of the sound field at the considered distances. In the emerging minima of the interference pattern (see Figures 6–8), intensity fluctuations were maximum. Naturally, when considering the incoherent sum of modes in Equation (5), the scintillation index corresponding to curves 1 and 2 in Figure 9 would not contain local maxima shown in Figure 12. The intensity scintillations averaged over the interference scale take the form of curves 3 in Figure 12 for $z = z_0 = 36$ m.

In conclusion, we note that the statistical modeling performed in this work was based on the algorithms developed by the authors in [28–30,38]. Numerical results were obtained using program codes developed by the authors (see a brief description in [24]) and implemented in the MATLAB environment.

## 6. Discussion

In this paper, we studied the decay laws for the average intensity of a low-frequency acoustic signal (sound frequency $f \sim$ hundreds of hertz) propagating in a shallow-water, two-dimensional, randomly inhomogeneous waveguide over distances ~10 km, which is typical in the study of a shallow sea. Of interest was a comparative analysis of the statistical effect of both fluctuations in the speed of sound in the sedimentary layers of the seabed and the random roughness of the water–bottom sediment interface on the energy loss of the signal in the course of propagation. Poorly studied scenarios with a bottom boundary of a shallow-water waveguide that is highly penetrable to signals seemed to be especially relevant for modeling. Such situations often occur in the water areas of seas with gas saturation in non-consolidated and weakly consolidated bottom sediments [8,21,37]. One example of this is the Arctic shelf zones, which are characterized by a variety of non-consolidated sediment properties, including elevated and spatially randomized gas content in sediments. In addition, in these regions, a distinctive feature

is the quasi-homogeneous stratification of the sound speed in the water column, which practically excludes volumetric random inhomogeneities of the sound speed (associated with the presence of internal waves) from the consideration. In other areas of the sea shelf, internal waves, as a rule, are the dominant factor of random perturbations and mask the influence of other inhomogeneities. Despite the extensive literature on the effect of both surface and volumetric random inhomogeneities in a waveguide on a propagating acoustic signal, it turned out that for the frequency range $f < 1$ kHz, there is no clear answer with quantitative estimates to the questions posed about the influence of different types of inhomogeneities on the energy losses of a sound signal and the specifics of this influence. In this work, on the basis of numerical statistical modeling, fundamental answers to these questions were obtained.

## 7. Conclusions

In the course of the study, the following results, which are novel, were obtained:

1.   It has been established that in shallow sea conditions with a highly penetrable (on average) water–bottom interface, the random roughness of the bottom boundary in the low-frequency range (hundreds of hertz and below) can be neglected in terms of signal intensity (average propagation loss). Of course, this statement does not apply to the subtle effects of changes in the interference structure and correlation functions of individual modes, modeled, for example, in [16]. From the point of view of the energy losses of a signal propagating in a shallow sea, the effect of volumetric random sound velocity inhomogeneities present in the underlying bottom sediments (similarly, in the water column) is much more pronounced. The greatest effect of the influence of a penetrable rough boundary on the average intensity, which was obtained for the range 200 Hz $< f < 500$ Hz, is $\approx 1$ dB at a distance of 10–15 km from the source, while sound velocity fluctuations in sediments (random impedance) in the same waveguide scenario can result in an effect of 15 dB or more.

2.   It is important to emphasize the different nature of the influence of the rough boundary and random bottom sediments on the signal intensity. If sound velocity fluctuations in sediments have an obvious distance-accumulating effect on signal losses (reductions in these losses, simultaneously leading to an increase in intensity fluctuations and, as a result, fast signal stochastization in the waveguide), then for random bathymetry, the effect of accumulation at the considered distances is almost absent. The latter is also confirmed by the behavior of the scintillation index.

3.   In terms of the modes propagating in the waveguide, the boundary inhomogeneities have a much stronger effect on higher modes with steeper grazing angles, which decay more strongly in the course of propagation. For the volumetric inhomogeneities of the speed of sound, a more uniform influence on all modes that form the field in the wave zone of the source (both on the first and higher modes) is characteristic.

4.   The difference between the influence of the rough boundary and random bottom sediments on the signal intensity is also manifested in the fact that the dependence on the characteristic scales of fluctuations is directly opposite. For inhomogeneities of the boundary interface, the effect increases with decreasing correlation radius (the slopes of the boundary locally increase), while the stronger the effects of volumetric sound velocity inhomogeneities, the larger the correlation scale is [26,27].

5.   The influence of interface inhomogeneities almost does not change with increasing signal frequency in the range $f < 1$ kHz (changes in the multimode interference pattern are of no interest in terms of averaged intensity laws). At the same time, the influence of volumetric fluctuations in the speed of sound increases in proportion to the frequency.

6.   If the water–sediment interface ceases to be highly penetrable (the condition $c \approx c_1$ is violated), then the effect of random sound velocity inhomogeneities in sediments on the signal transmission loss decreases. At the same time, an increase in the "rigidness" of the boundary, or its "softness", does not lead to a fundamental change in the

influence of its random roughness on the decrease in intensity and the stochastization of the signal in the waveguide.

7. Since the effect of random boundary roughness is rather small for the waveguide scenarios considered and the frequency range $f < 1$ kHz, it seems reasonable to use an approximate approach (the perturbation method in terms of eigenvalues and eigenfunctions) to describe the effect of two-dimensional random inhomogeneities in a shallow sea, developed in works [39–42]. At the same time, it should be kept in mind that in the presence of strong fluctuations in the waveguide caused by volumetric inhomogeneities in the speed of sound, this method is not suitable [24–27].

The patterns of behavior of the average intensity and its fluctuations, which describe the signal transmission loss in the random environment waveguide–interface–bottom sediments, revealed in this work, are of interest from a fundamental point of view. Statistical modeling allows a more detailed understanding of the physical picture of the considered phenomena than can be conducted on the basis of approximate theoretical methods of analysis [1–12]. The obtained results regarding the influence of the considered inhomogeneities, their physical analysis, and quantitative estimates can be useful in practical terms for predicting the transmission loss (and fluctuations) of low-frequency signals in shallow Arctic regions and marine areas with similar conditions. Such a forecast is necessary when solving problems of underwater detection, communications, and the exploration of minerals on the shelf of the Arctic seas. An important private problem of ecology is the problem of reducing the impact of anthropogenic signals and noise on marine mammals living on the shelf of the Arctic seas [8,21].

Undoubtedly promising, from our point of view, are studies that generalize the results of statistical modeling obtained in this work to three-dimensionally inhomogeneous waveguides. Such waveguides can have, in particular, non-planar interfaces and sound velocity fluctuations in the horizontal plane [43]. The modeling method used in this work and described earlier in [24–29,34,44] in many important cases allows one to study the influence of three-dimensional random inhomogeneities in question.

**Author Contributions:** Conceptualization, O.E.G. and I.O.Y.; methodology, O.E.G. and I.O.Y.; software, O.E.G. and I.O.Y.; validation, O.E.G.; formal analysis, O.E.G.; investigation, O.E.G. and I.O.Y.; writing—original draft preparation, O.E.G.; writing—review and editing, O.E.G. and I.O.Y.; visualization, O.E.G.; supervision, I.O.Y.; project administration, I.O.Y. All authors have read and agreed to the published version of the manuscript.

**Funding:** This research was funded by project "Development of a climate monitoring system for the Far Eastern seas", which is a part of the overall program "Monitoring of key areas of the World Ocean, coastal zones and seas of Russia, including characteristics of circulation, heat content and greenhouse gas and energy flows at the ocean-atmosphere interface". The research was also carried out as a part of the Russian State assignment on the topic "Study of the fundamental basis of the origin, development, transformation, and interaction of hydroacoustical, hydrophysical, and geophysical fields of the World Ocean" (state number registration: AAAA-A20-120021990003-3).

**Institutional Review Board Statement:** Not applicable.

**Informed Consent Statement:** Not applicable.

**Data Availability Statement:** Not applicable.

**Conflicts of Interest:** The authors declare no conflict of interest.

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
