# Peer review of "On Average Losses of Low-Frequency Sound in a Two-Dimensional Shallow-Water Random Waveguide"

_jmse, doi:10.3390/jmse10060822_

Round 1

Reviewer 1 Report

On average losses of low frequency sound in course of propagation in a two-dimensional waveguide with random bottom and rough penetrable bottom boundary is presented in the research. To improve the quality of the publication following are the suggestions:

The title of the article is too long, should be reduced.

The division of the abstract should be in the form of: (a) research problem, (b) research methodology, (c) results and findings and (d) conclusion and recommendations.

The introduction section does not have any paragraphs. Suitable breaks should be provided at the end of each statement.

In Section 3, Stochastic Waveguide Model, no model is presented.

Figures have too lengthy captions. It is suggested to use small captions with the figures, however the description of the figures should be provided in the text.

If the results are presented in the Section 4, the title of the section should reflect the information.

Discussion and Conclusion should be separated, precise conclusion should be provided at the end of the article under separate heading.

Reviewer 2 Report

In this paper, the authors study the decay laws for the average intensity of a low-frequency acoustic signal,  the influence of random bathymetry is considered based on the local-mode approach and statistical modeling by first order evolution equations. It is very interesting and well organized. However, it has some problems that need to be solved further. 

1、The formula between lines 145-146 is very complex and has no label. The authors need to explain the meaning of each symbol in detail.

2、Mathematical Statement of the Problem and Some Analytics, Stochastic Waveguide Model and Statistical Analysis of the Propagation Loss are described separately. The authors need to logically explain the relationship between them and reasonably arrange each section.

3、Many Figures have no legend to distinguish the meaning of different linearity, such as Figure 1, 2, and so on.

4、 Discussion and Conclusions are together, which is not clear enough. It is suggested that the authors take the Conclusions as an independent Section and add some key experimental data and results.

5、The innovation of the article is not well reflected in the abstract and conclusion. How can the authors ensure that the proposed model can be applied to complex marine environment?

In order to qualify for publication in JMSE, the article must be improved according to the comments to the authors.

Round 2

Reviewer 1 Report

Most of the points have been addressed by the authors, but one of the main concern i.e., Stochastic Waveguide Model is still incomplete. The authors are referring some equation from the introduction section and some figures from the result section. The main work is still weak and the model should be clearly explained in this section, either move some equations from the above section or use graphical representation. In current form it seems that the actual contribution is too less and should be presented clearly. 

Reviewer 2 Report

Thank the authors for their efforts. The authors have adequately addressed all my concerns in the review, and did a good job to revise and improve the paper. The paper now is suitable for publication in JMSE in its current form.

Author Response

The authors are grateful to the reviewer for the time spent on reviewing the manuscript and comments that contributed to the improvement of the work. Many thanks